# Cancer cell-expressed SLAMF7 is not required for CD47-mediated phagocytosis

Yuan He[1], Renee Bouwstra[1], Valerie R. Wiersma[1], Mathilde de Jong[1], Harm Jan Lourens[1], Rudolf Fehrmann[2], Marco de Bruyn[3], Emanuele Ammatuna[1], Gerwin Huls[1], Tom van Meerten[1] & Edwin Bremer[1]

CD47 is a prominent new target in cancer immunotherapy, with antagonistic antibodies currently being evaluated in clinical trials. For effective evaluation of this strategy it is crucial to identify which patients are suited for CD47-targeted therapy. In this respect, expression of the pro-phagocytic signal SLAMF7 on both macrophages and cancer cells was recently reported to be a requisite for CD47 antibody-mediated phagocytosis. Here, we demonstrate that in fact SLAMF7 expression on cancer cells is not required and does not impact on CD47 antibody therapy. Moreover, SLAMF7 also does not impact on phagocytosis induction by CD20 antibody rituximab nor associates with overall survival of Diffuse Large B-Cell Lymphoma patients. In contrast, expression of CD47 negatively impacts on overall and progression free survival. In conclusion, cancer cell expression of SLAMF7 is not required for phagocytosis and, in contrast to CD47 expression, should not be used as selection criterion for CD47-targeted therapy.

[1] Department of Hematology, University of Groningen, University Medical Center Groningen (UMCG), Groningen, GZ 9713, The Netherlands. [2] Department of Medical Oncology, University of Groningen, University Medical Center Groningen (UMCG), Groningen, GZ 9713, The Netherlands. [3] Department of Gynecological Oncology, University of Groningen, University Medical Center Groningen (UMCG), Groningen, GZ 9713, The Netherlands. Correspondence and requests for materials should be addressed to T.v.M. (email: t.van.meerten@umcg.nl) or to E.B. (email: e.bremer@umcg.nl)

The CD47/SIRP-α axis has been established as an important regulator of innate anti-cancer immunity, with many if not all malignancies overexpressing the receptor CD47 that binds to phagocyte-expressed SIRP-α[1–3]. CD47-mediated triggering of SIRP-α inhibits phagocytic removal of cancer cells and reduces the immunogenic processing of cancer cells by macrophages and dendritic cells[2,4,5]. Consequently, both innate and adaptive anticancer immunity is suppressed. Correspondingly, high CD47 expression is associated with poor clinical prognosis in various malignancies[6,7].

CD47/SIRP-α-blocking antibodies enhance antibody-dependent cellular phagocytosis (ADCP) of cancer cells upon co-treatment with anticancer monoclonal antibodies[6,8]. For instance, co-treatment of anti-CD20 antibody rituximab with the CD47-blocking murine antibody B6H12 synergized the phagocytic elimination of xenografted human CD20pos non-Hodgkin lymphoma (NHL) cancer cells in murine models in the absence of noticeable toxicity[6]. Correspondingly, humanized CD47-blocking antibodies are currently being evaluated in phase-1 clinical trials (NCT02216409/NCT02367196). Thus, CD47 is a prominent new target in cancer immunotherapy, particularly in B-cell malignancies, in which e.g. combination of a CD47 antibody with the CD20 antibody rituximab is being explored in clinical trials.

However, several reports have highlighted potential immunoregulatory proteins that may impact on the efficacy of CD47-targeted therapy[9–11]. For instance, expression of LILRB1 on macrophages inhibited induction of cancer cell phagocytosis by a CD47-blocking antibody by direct binding to MHC class I and inhibition of macrophage activity, which was reversed by antibody-mediated blocking of LILRB1[11]. Further, it was recently reported that the expression of the pro-phagocytic receptor SLAMF7 on macrophages and cancer cells was required for phagocytosis induction upon treatment with a CD47 blocking therapeutic antibody[10]. Specifically, macrophages obtained from SLAMF7 knock-out mice proved to be defective in phagocytosis of cancer cells. Further, SLAMF7 expression on hematopoietic cancer cells was reported as a requisite for phagocytosis upon treatment with a CD47 blocking antibody. The premise arising from this finding is that only hematopoietic cancers that express high levels of SLAMF7 are suitable targets for CD47 blocking therapy. As such, diffuse large B-cell lymphoma (DLBCL), the most common subtype of non-Hodgkin's lymphoma (NHL), was identified as a suitable target for CD47 blocking therapy based on its high SLAMF7 mRNA levels.

In the current report, we aimed to further delineate the potential role of SLAMF7 expression on cancer cells for CD47-targeted and monoclonal antibody-based therapy in DLBCL. Surprisingly, we found that surface expression of SLAMF7 is not required for phagocytosis of DLBCL cells and does not correlate with phagocytosis by CD47 blocking antibody treatment. Similarly, phagocytosis induction upon treatment with CD20 antibody rituximab alone or in combination with CD47 antibody does not correlate with, nor requires, cancer cell surface expression of SLAMF7. Correspondingly, SLAMF7 mRNA expression does not correlate with overall survival (OS) after R-CHOP treatment in a large transcriptomic dataset of gene expression profiles (GEP) of 680 DLBCL patients, whereas expression of CD47 does. Taken together, expression of SLAMF7 is not required nor impacts on phagocytosis upon CD47 antibody treatment and should not be used as a selection criterion for CD47-targeted antibody therapy. Rather, our data indicate that the expression level of CD47 itself may be a primary selection criterion in DLBCL.

## Results

**CD47-mediated phagocytosis does not require SLAMF7 on DLBCL.** Since SLAMF7 was postulated to be critical for CD47 antibody-mediated phagocytosis and DLBCL was postulated to be a prime target for CD47 antibody therapy, we first determined expression of SLAMF7 in DLBCL cell lines and primary DLBCL cells and found surface expression of SLAMF7 in only 1 of the 7 DLBCL cell lines tested (Fig. 1a, b), with mRNA for SLAMF7 being detected only in 2 out of 7 cell lines (Fig. 1c). Importantly, surface expression of SLAMF7 was also not detected on primary patient-derived DLBCL or mantle cell lymphoma (MCL) cells (Fig. 1d). In contrast, expression of high levels of SLAMF7 was detected on the surface of macrophages, including on primary macrophages obtained from DBLCL and MCL patients (Fig. 1e, f). To investigate if tumor-expressed SLAMF7 was a requisite for phagocytosis of DLBCL cells upon CD47 targeting, we generated type 1 macrophages (MØ) as the prototype pro-inflammatory macrophage subtype associated with anti-cancer activity. These macrophages were mixed with fluorescently labeled DBLCL cells and phagocytosis upon CD47 targeting was assessed. Importantly, despite the absence of cancer cell-expressed SLAMF7, CD47 targeting significantly induced phagocytosis of 7 out of 7 fluorescently (V450)-labeled DLBCL lines by macrophages when using F(ab′)2 fragments (Fig. 2a and Supplementary Movies 1 and 2). Of note, F(ab′)2 fragments were used for this analysis in order to delineate the impact of CD47 blocking in the absence of potential Fc/FcR-mediated effects that may occur when using full antibodies. To further validate engulfment of tumor cells, cells were stained with either V450 or PHrodo-Red dye, a dye only emitting fluorescence after internalization. In both settings, phagocytosis of DLBCL cells in CD11b-stained macrophages was detected after treatment with CD47 F(ab)′2 (Fig. 2b). Assessment of phagocytosis using flow cytometry yielded similar results (Fig. 2c), with significant phagocytic uptake of six out of seven DLBCL cell lines (Fig. 2d). Of note, staining for caspase-3-positive apoptotic cells, using IncuCyte caspase-red staining, identified that CD47 F(ab)′2 treatment triggered phagocytosis of viable cells (Supplementary Figure 1A). Moreover, in the 2 h time-line of this assay no additional caspase-positive (and/or fragmented caspase-negative) apoptotic bodies were detected (data not shown). Thus, CD47 blockade triggered phagocytosis of viable DLBCL cells.

It is well-established that various macrophage subtypes can be detected in the tumor micro-environment (TME), with the predominant focus in literature being on the so-called M2 subtype that is thought to contribute to the immunosuppressive milieu in the tumor[12]. However, also non-polarized M0 as well as pro-inflammatory M1 polarized macrophages have been reported in the TME of various cancers[12]. Correspondingly, we used CIBERSORT[13,14] to estimate the fraction of different macrophage subtypes in the DLBCL micro-environment in a large gene expression database of DLBCL (Fig. 2e). Therefore, the impact of targeting of CD47 using F(ab′)2 fragments was also assessed for M0 and M2 macrophages, with increased phagocytosis of DLBCL cells upon treatment being detected for macrophages differentiated into M1 (LPS/IFN-γ), M2 (IL-10), and M0 (M-CSF/GM-CSF) macrophages (Fig. 2f, g). Thus, for all of the macrophage subtypes found in the TME of DLBCL, SLAMF7 expression on cancer cells is not required for anticancer activity upon CD47 targeting, with M2 macrophages appearing to respond more effectively to CD47-targeting than M1 macrophages while having lower SLAMF7 expression (Supplementary Figure 1B, C).

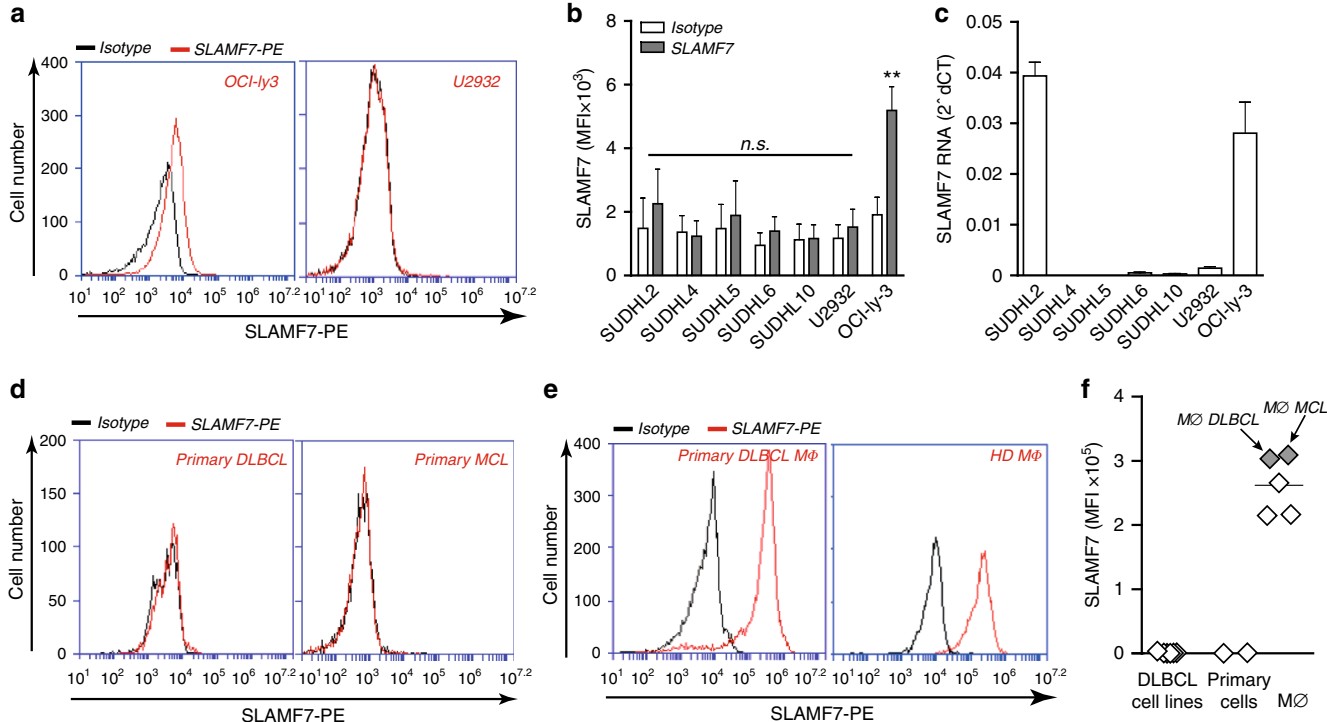

**Fig. 1** SLAMF7 expresses on primary macrophages but not on DLBCL cells. **a** Representative graph of SLAMF7 expression in two DLBCL lines (SLAMF7 positive in OCI-ly3 cells, SLAMF7 negative in U2932 cells). **b** Surface expression levels of SLAMF7 in a panel of seven DBLCL cell lines ($n = 3$). **c** mRNA expression levels of SLAMF7 in a panel of seven DLBCL cell lines ($n = 3$). **d** Expression of SLAMF7 on primary patient-derived DLBCL cells. **e** Representative graph of SLAMF7 expression on macrophages from the DLBCL patient or healthy donors. **f** Quantification of SLAMF7 expression in DLBCL lines, primary patient-derived DLBCL cells, macrophages from healthy donors and macrophages from patients with B-cell malignancies. Error bars stand for standard deviation (SD)

**SLAMF7 does not correlate with CD47-mediated phagocytosis**. Next, we evaluated whether phagocytic activity of a clinically relevant CD47 targeting antibody might similarly be independent of SLAMF7 expression on cancer cells. Hereto, we used the antibody Inhibrix, an IgG4 containing antibody currently being evaluated in clinical trials for B cell malignancies including DBLCL (NCT02953509). Use of an IgG4 domain limits unwanted Fc/FcR-interactions and should largely restrict activity of the antibody to blocking of CD47/SIRPα interaction. Importantly, treatment with this clinically relevant CD47 antibody was comparable in efficacy to that of treatment with the CD47 F(ab')2 (Fig. 3a). Moreover, inhibrix also effectively induced phagocytosis upon treatment of SLAMF7-negative primary patient-derived DLBCL and MCL cells by autologous patient-derived macrophages, yielding significant increases in phagocytosis of ~15% and 8%, respectively (Fig. 3b). Thus, the effect of a clinically relevant CD47 blocking IgG4 antibody is mediated by blocking of the SIRPα/CD47 interaction and does not require expression of SLAMF7 on cancer cells.

To further investigate the potential relevance of cancer-expressed SLAMF7, other B cell NHL cell lines cells with varying levels of SLAMF7 expression were evaluated for phagocytosis upon CD47-targeting. Specifically, the NHL cell line Raji, BJAB, and Z138 significantly expressed cell surface SLAMF7 (Fig. 3c), whereas Daudi and Ramos had weak and non-significant expression of SLAMF7 (Fig. 3c). Nevertheless, all of these cell lines were significantly phagocytosed upon treatment with CD47 F(ab')2 irrespective of expression of SLAMF7 (Fig. 3d). Correspondingly, expression of SLAMF7 did not at all correlate with the level of experimental phagocytosis induced by treatment with CD47 F(ab')2 (Fig. 3e, $r^2 = 0.00012$), nor with CD47 antibody Inhibrix (Fig. 3f, $r^2 = 0.0016$). Taken together, these data clearly demonstrate that expression of SLAMF7 on hematopoietic cancer cells is not required for phagocytosis by macrophages upon CD47 blocking therapy. Similarly, SLAMF7 expression (or lack thereof) did not correlate with the extent of phagocytosis induced upon treatment of a B-NHL cell line panel with CD20 antibody rituximab alone (Fig. 3g) or upon combination treatment with rituximab and inhibrix (Fig. 3h, i). Indeed, combined treatment failed to reach a statistically significant beneficial effect in the SLAMF7-positive cell line Oci-Ly3, whereas significant improvement was detected in the SLAMF7-negative cell lines (Fig. 3i). Thus, in B-NHL cell lines the expression of SLAMF7 is not required for induction of phagocytosis by CD47 antibodies, nor for phagocytosis induction upon treatment with rituximab.

**SLAMF7 mRNA does not impact survival in R-CHOP-treated DBLCL**. Next, we evaluated the potential clinical impact of SLAMF7 on CD47 blocking, specifically in the context of combination with the clinically relevant antibody rituximab, which is part of the standard-of-care treatment for DLBCL patients. Of note, combination of rituximab with CD47 blocking antibodies is currently being investigated in several phase I clinical trials (NCT02367196; NCT02953509). In a large transcriptomic dataset of GEP of 680 DLBCL patients that were treated with R-CHOP (Supplementary Tables 1 and 2), the OS of patients with high SLAMF7 expression did not differ from that of patients with low expression of SLAMF7 (Fig. 4a, $p = 0.2$). In a similar analysis with CD47, patients with high mRNA expression of CD47 did have a significantly worse OS than patients with low expression of CD47 (Fig. 4b, $p = 0.0009$). Further, when CD47 expression was analyzed within the SLAMF7 high and low population, no significant impact of SLAMF7 on OS could be detected (Fig. 4c). Similarly,

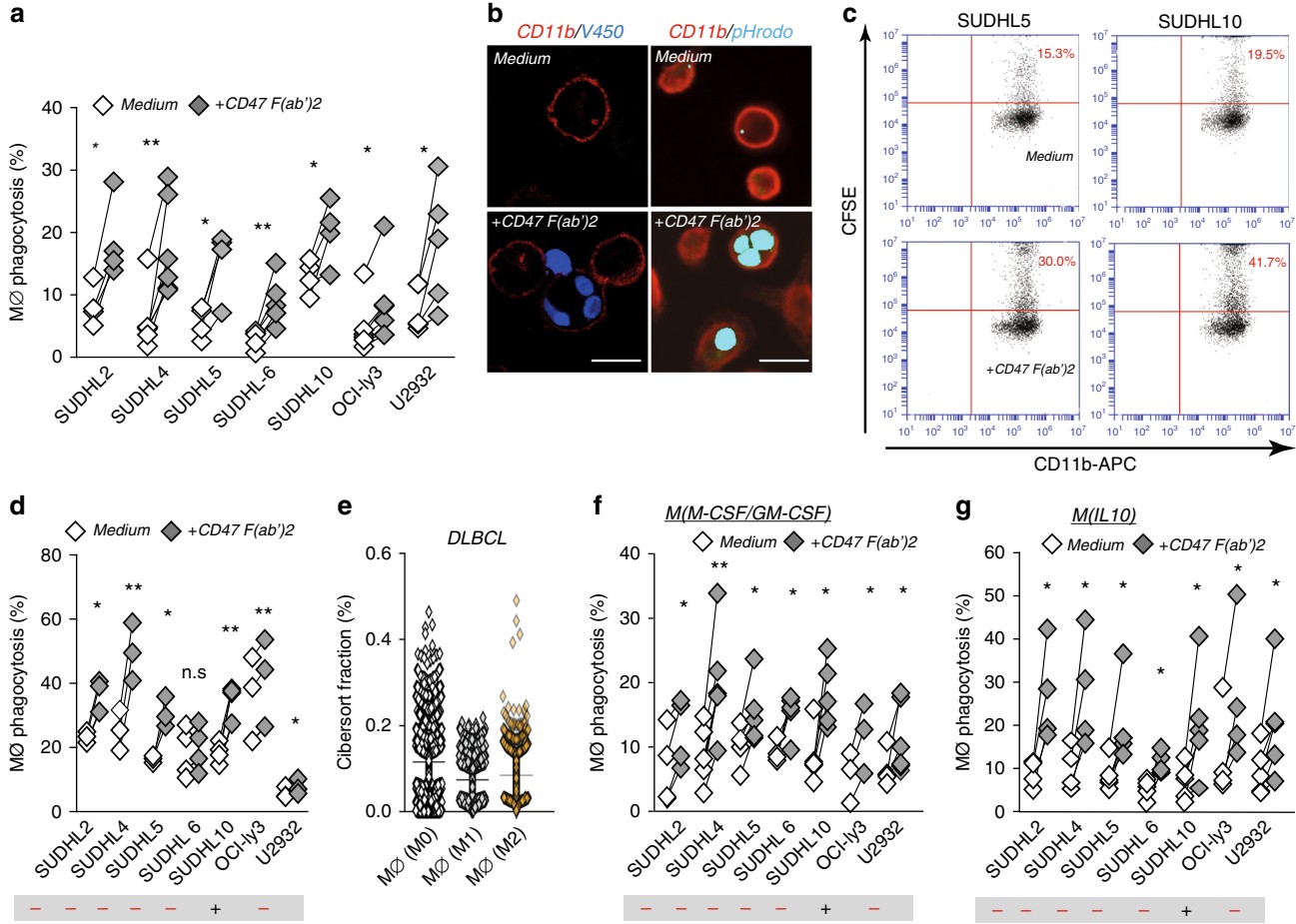

**Fig. 2** Tumor-expressing SLAMF7 is not required for induction of phagocytosis upon CD47-targeting treatment in DLBCL cells. **a** Percentage of phagocytosis of DLBCL cell lines by allogeneic human macrophages primed with LPS/IFN-γ upon 3 h treatment with F(ab')2 of anti-CD47 antibody inhibrix (CD47 F(ab')2) vs. untreated cells ($n = 3-5$). **b** Representative microscopy pictures of phagocytosis of tumor cells by macrophages primed with LPS/IFN-γ upon 3 h treatment with CD47 F(ab')2 (left, MØ + V450-labeled OCIly3 cells, right, MØ + pHrodogreen-labeled SUDHL5 cells). Scale bar = 20 μm. **c** Representative graphs of flow cytometric analysis for phagocytosis of tumor cells by macrophages with LPS/IFN-γ upon 3 h treatment with CD47 F(ab')2 (left, MØ + SUDHL5, right, MØ + SUDHL10). **d** Quantification of phagocytosis of DLBCL cell lines by flow cytometric analysis. Experimental setting is the same as in (**a**) ($n = 3-4$). **e** Percentage of different types of macrophages from cibersort fraction of DLBCL biopsies ($n = 1804$). **f** Percentage of phagocytosis of DLBCL cell lines by allogeneic type 0 human macrophages upon 3 h treatment with F(ab')2 of anti-CD47 antibody inhibrix (CD47 F(ab')2) vs. untreated cells ($n = 4-6$). **g** Percentage of phagocytosis of DLBCL cell lines by allogeneic human macrophages primed with IL-10 upon 3 h treatment with F(ab')2 of anti-CD47 antibody inhibrix (CD47 F(ab')2) vs. untreated cells ($n = 4-6$). Statistics was performed using paired Student's t-test. n.s. = not significant, $*p < 0.05$; $**p < 0.01$; $***p < 0.001$; $****p < 0.0001$. Error bars stand for standard deviation (SD)

when SLAMF7 expression was analyzed within the CD47 high and CD47 low quartiles, high expression of CD47 associated with poor survival, but was not further impacted by high or low expression of SLAMF7 (Fig. 4d). Thus, SLAMF7 does not impact on treatment outcome in a large cohort of R-CHOP-treated patients. Of note, the outcome of this SLAMF7 GEP analysis should in future studies be confirmed with a similar analysis of SLAMF7 at the protein level, particularly as we did not detect SLAMF7 on the primary DLBCL sample (Fig. 1d).

## Discussion
The data presented here demonstrate that surface expression of SLAMF7 on hematopoietic cancer cells, specifically on B cell malignant cells, is not required for phagocytosis upon CD47 blocking treatment, nor upon combination treatment with rituximab. Further, mRNA expression of SLAMF7 is not predictive for survival in a large cohort of R-CHOP-treated DLBCL patients, whereas mRNA expression of CD47 is predictive. The important corollary of these findings is that cancer cell expression

of SLAMF7 does not associate with or predict for therapeutic effects of CD47-targeting drugs. As such, expression of SLAMF7 should not be used as an inclusion/exclusion criterion for clinical trials that evaluate CD47 targeting.

The findings presented here clearly contrast to the conclusions arrived at by Chen et al., with their conclusion being that SLAMF7 expression on both macrophages and tumor cells is a requisite for phagocytosis upon CD47 antibody treatment both in vitro and in vivo[10]. However, only two B-cell lines were presented as examples of susceptible SLAMF7-positive cells, with DLBCL being proposed as a suitable target for CD47 blocking therapy solely based on high SLAMF7 mRNA levels in a patient cohort. Using the same antibody clone as Chen et al., only 1 out of 7 DLBCL cell lines was found to detectably express cell surface SLAMF7. Notably, also no SLAMF7 surface expression was observed on primary patient-derived leukemic DLBCL and MCL cells. Moreover, the F(ab')2 of CD47 antibody inhibrix as well as the full antibody inhibrix-induced significant phagocytosis in all these DLBCL lines. In line with this, there was no correlation between SLAMF7 expression and phagocytosis by CD47

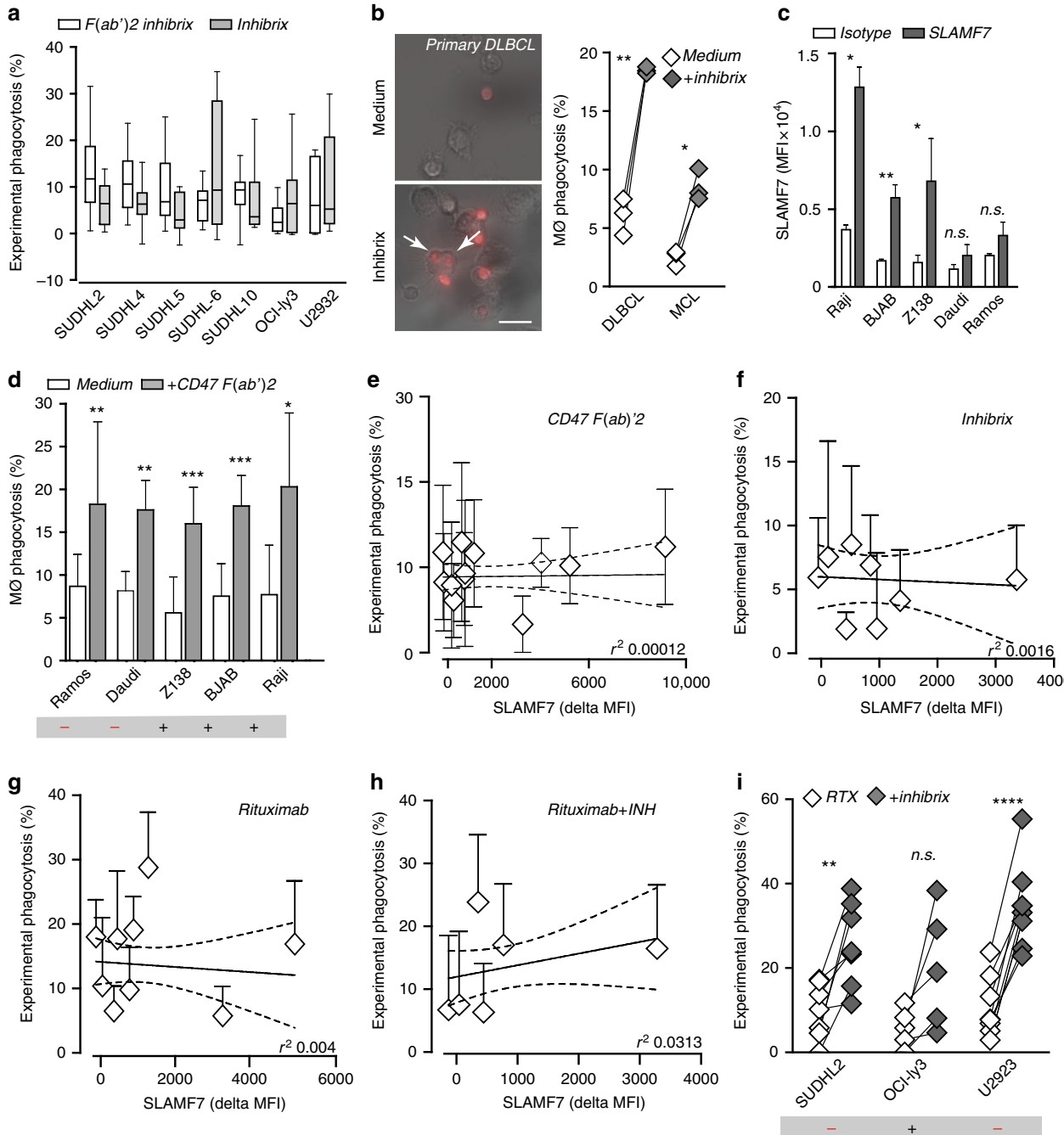

**Fig. 3** Efficacy of CD47-targeting antibodies in B-cell malignant cells does not correlate with SLAMF7 expression. **a** Experimental phagocytosis of DLBCL lines by macrophages either upon CD47 F(ab')2 treatment or inhibrix treatment ($n = 3$). Box plot contains center line representing median and whiskers representing 5–95%. **b** Representative microscopy pictures of phagocytosis of primary DLBCL cells by autologous macrophages upon 3 h treatment with inhibrix. Quantification of phagocytosis of primary DLBCL and MCL cells by autologous macrophages. **c** Quantification of surface SLAMF7 expression on five NHL lines ($n = 3$). **d** Percentage of phagocytosis of NHL cell lines by allogeneic human macrophages primed with LPS/IFN-γ upon 3 h treatment with F (ab')2 of anti-CD47 antibody inhibrix (CD47 F(ab')2) vs. untreated cells ($n = 3–4$). **e** Correlation between SLAMF7 expression and the percentage of experimental phagocytosis induced by CD47 F(ab')2 in NHL and DLBCL cell panel ($n = 3$). **f** Correlation between SLAMF7 expression and the percentage of experimental phagocytosis induced by anti-CD47 antibody inhibrix in DLBCL cell panel ($n = 3–4$). **g** Correlation between SLAMF7 expression and the percentage of experimental phagocytosis induced by Rituximab in NHL and DLBCL cell panel ($n = 3$). **h** Correlation between SLAMF7 expression and the percentage of experimental phagocytosis induced by the combinatory treatment of Rituximab and Inhibrix in DLBCL cell panel ($n = 3$). **i** Experimental phagocytosis of tumor cells by macrophages upon RTX treatment or combination treatment with inhibrix ($n = 3$). Experiments with primary patient-derived samples were performed in triplicates. Statistics was performed using paired Student's $t$-test. n.s. = not significant, $*p < 0.05$; $**p < 0.01$; $***p < 0.001$; $****p < 0.0001$. Error bars stand for standard deviation (SD)

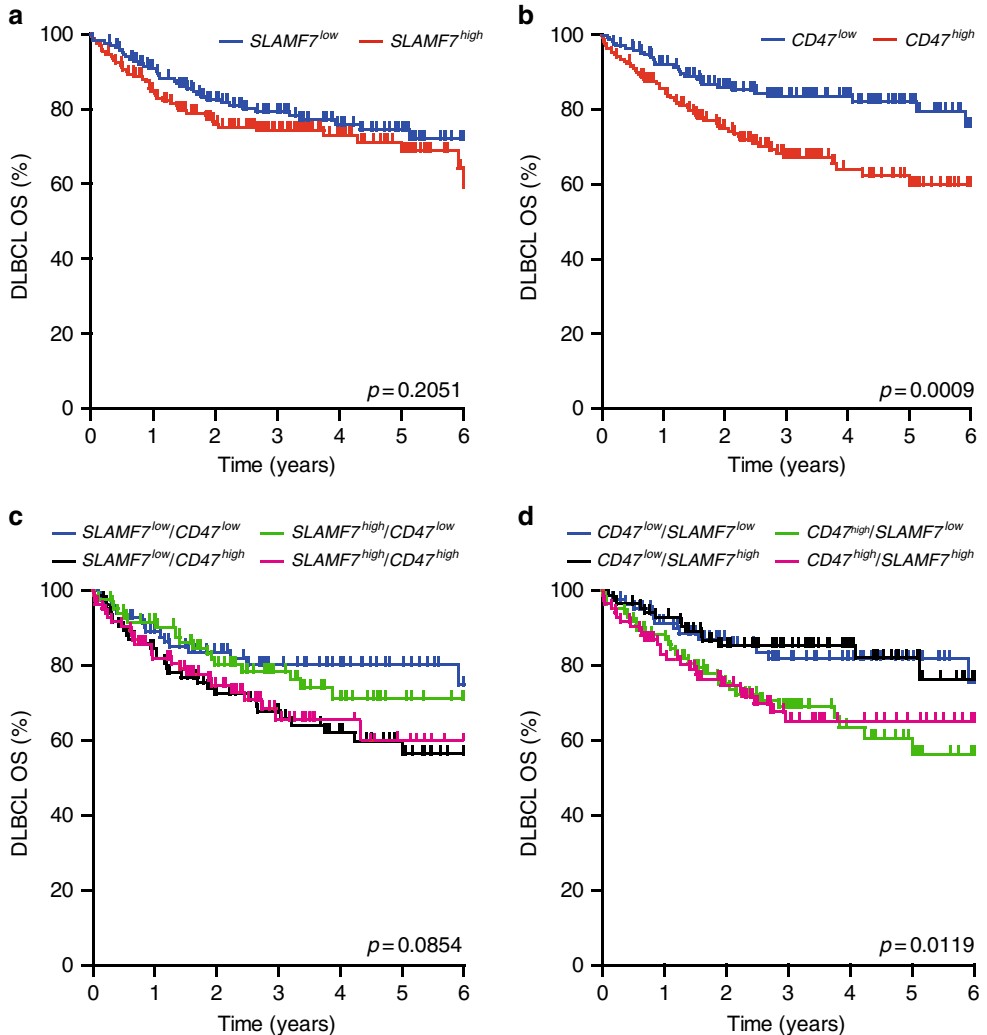

**Fig. 4** mRNA expression of SLAMF7 does not, but of CD47 does, associate with survival in DLBCL patients. **a** Kaplan–Meijer curve analysis of survival of DLBCL patients within high and low SLAMF7-expressing quartiles. **b** Kaplan–Meijer curve analysis of survival of DLBCL patients within high and low CD47-expressing quartiles. **c** Kaplan–Meijer curve analysis of survival of DLBCL patients within high and low SLAMF7-expressing quartiles additionally sorted on high and low expression of CD47. **d** Kaplan–Meijer curve analysis of survival of DLBCL patients within high and low CD47-expressing quartiles additionally sorted on high and low expression of SLAMF7

antibodies treatment in a cohort of DLBCL and NHL lines ($r^2 =$ 0.00012). Thus, expression of SLAMF7 on cancer cells is not a requisite for phagocytosis upon CD47 antibody treatment.

Expression of SLAMF7 on cancer cells also did not impact on the in vitro phagocytic activity of macrophages upon treatment with the CD20 antibody rituximab alone, or in combination with CD47 antibody inhibrix. These data correspond to the lack of association of SLAMF7 mRNA expression with patient survival after rituximab and CHOP treatment as evaluated in a large cohort of 680 DLBCL patients. Of note, in this cohort, expression of *CD47* did associate with OS in R-CHOP-treated patients, as patients that express high CD47 have a worse outcome. Thus, these data suggest that the therapeutic effect of rituximab may potentially be augmented by co-treatment with CD47 blocking antibody. In contrast, the data provided here do not support SLAMF7 as a potential selection marker for response, nor do these data suggest that combination of rituximab with, e.g. SLAMF7 targeting antibodies, such as elotuzumab, would be a potential combinatorial approach to augment rituximab activity given the lack of expression in the majority of cell lines tested here. In this respect, combination of rituximab with CD47 antibody targeting is being clinically evaluated for patients with

relapsed/refractory B-cell non-Hodgkin's lymphoma (NCT02953509). Of note, although for DLBCL the expression of SLAMF7 does not impact on phagocytosis upon CD47-targeting treatment, its impact especially in multiple myeloma (MM) may well be different. Indeed, surface expression of SLAMF7 is well-established on both normal plasma cells and MM and SLAMF7 has been exploited as tumor target in MM using elotuzumab[15]. Indeed, heavily pre-treated MM patients who received a combination of elotuzumab, lenalidomide, and dexamethasone had a significant prolonged progression free survival compared to lenalidomide and dexamethasone alone[16].

Expression of SLAMF7 on macrophages was reported to be a requisite for phagocytosis upon CD47-targeting therapy, as SLAMF7 knock-out in murine macrophages abrogated phagocytosis upon CD47 targeting. Further, co-treatment with a blocking SLAMF7 antibody inhibited CD47-mediated phagocytosis of cancer cells by human macrophages[10]. Interestingly, in a transcriptome analysis of polarized macrophages the expression of SLAMF7 was found to be clearly associated with the M1 polarization status of macrophages[17]. In line with this, expression of SLAMF7 was higher on M1-than on M2-polarized macrophages in our study (Supplementary Figure 1A). Although

pro-inflammatory M1 polarized macrophages have been reported in the TME of various cancers, the predominant focus in literature is on the so-called M2 subtype that is thought to particularly contribute to the immunosuppressive milieu[12]. In our experiments, M2 macrophages proved to be equally effective or perhaps slight more effective than M1 macrophages in triggering CD47 antibody-dependent phagocytosis of DBLCL cells (Supplementary Figure 1B). This finding is in agreement with a report that M2 macrophages more effectively phagocytosed rituximab-opsonized tumor cells than M1 macrophages[18]. Taken together, these data suggest that the expression level of SLAMF7 on macrophages may not impact on macrophage activity after CD47 targeting. Thus, it will be of interest to further determine whether SLAMF7 expression on human M2-polarized and M1-polarized macrophages indeed is critical for phagocytosis, e.g. by knockdown studies in cord blood stem cell-derived macrophages.

In conclusion, mRNA and/or protein expression levels of SLAMF7 on hematopoietic cancer cells should not be used as selection/exclusion criterion for future clinical studies that evaluate the therapeutic potential of CD47-blockade or the combination with CD47 blocking therapy.

## Methods

**Reagents and antibodies.** PE-labeled anti-human SLAMF7 antibody (clone, 162.1) and PE-labeled isotype control were purchased from Biolegend. APC-labeled anti-CD3, FITC-labeled anti-CD19 and APC-labeled anti-CD47 were purchased from Immunotools (Germany). FITC-labeled anti-CD20 was purchased from Thermofisher. Alexa594-labeled CD11b was purchased from Biolegend. Anti-human CD47 IgG4 antibody (Inhibrix, clone B6H12) was generated in-house by Aduro Biotech Europe (ABE). F(ab')2 of Inhibrix was prepared with Pierce F(ab')2 preparation kit. F(ab')2 generation was confirmed by staining for human IgG4 (Supplementary Figure 1D). Cell proliferation dye V450 was purchased from Thermofisher. Phrodo Green pH indicators was purchased from Thermofisher. Lymphoprep was purchased from Axis-Shield PoC AS, Norway. pHrodo Green intracellular pH indicator was purchased from ThermoFisher. RNeasy mini kit was purchased from Qiagen. iScript cDNA Synthesis Kit was purchased from Bio-Rad.

**Cell lines and primary patient-derived B-cell malignancies.** Cell lines used in this study were obtained from the American Type Culture Collection (Manassas, VA) or the Deutsche Sammlung from Microorganimen und Zellculturen (Braunschweig, Germany) and cultured at 37 °C in humidified 5% $CO_2$ containing atmosphere. DLBCL cell lines SUDHL2, SUDHL5, SUDHL10, and OCI-ly3 were cultured in RPMI culture medium (Lonza) supplemented with 20% fetal calf serum (Thermo Scientific) in the present of glutamine (100 μM, Gibco). DLBCL cell lines SUDHL4, SUDHL6, and U2932 and NHL cell lines BJAB, Daudi, Ramos, Raji, and Z138 were cultured in RPMI 1640 culture medium supplemented with 10% fetal calf serum (Thermo Scientific). Peripheral blood mononuclear cells (PBMCs) from patients' blood were isolated by using gradient centrifugation with lymphoprep and phenotyped for CD3, CD19, CD20, CD47, and SLAMF7. This study was carried out in The Netherlands in accordance with International Ethical and Professional Guidelines (the Declaration of Helsinki and the International Conference on Harmonization Guidelines for Good Clinical Practice). The use of anonymous rest material is regulated under the code for good clinical practice in the Netherlands. Informed consent was waived in accordance with Dutch regulations.

**Preparation of primary human macrophages.** Monocytes were enriched from isolated PBMCs, obtained from healthy donors after informed consent, by MACS sorting using CD14 magnetic beads (Miltenyi Biotec). Monocytes were differentiated into macrophages (M0) in RPMI 1640 culture medium + 10% FCS supplemented with GM-CSF (50 ng/ml) and M-CSF (50 ng/ml) for 7 days. To generate type 1 macrophages, M0 cells were primed by LPS and IFN-γ for additional 24 h. To generate type 2 macrophages, M0 cells were primed by IL-10 for an additional 48 h. To generate cord blood-derived monocytes, CD34+ stem cells were isolated by MACS sorting using CD34 magnetic beads, followed by stem cell culture in the presence of cytokine mixture for 14 days[19]. Subsequently, monocytes were differentiated to type 0 macrophage as described above. To isolate patient-derived monocytes, PBMCs from patients were seeded into six-well plates for 3–4 h after which floating cells were removed. Monocytes were then gently washed with PBS (2–3 times) and cultured with fresh medium containing GM-CSF and M-CSF as described above.

**Surface expression of SLAMF7 on B-cell malignant cells.** Both malignant B-cells ($5 \times 10^5$/ml) and primary patient-derived blasts were stained with anti-SLAMF7 antibody (2.5 μg/ml) or isotype control on ice for 1 h. Subsequently, cells were washed with ice-cold PBS (three times) and resuspended in fresh medium. Cellular surface expression of SLAMF7 was then determined by flow cytometry (Accuri, BD).

**qRT-PCR analysis in DLBCL lines.** Cells were washed with cold PBS and then cell pellet was harvested by centrifugation at 5000 rpm for 15 min. Next, RNA was isolated from cell pellet by RNeasy mini Kit according to manufacturer's introductions, and subsequently cDNA was synthesized from quantified RNA with iScript cDNA Synthesis Kit according to manufacturer's introductions. qRT-PCR analysis for SLAMF7 was performed on a Bio-Rad thermal cycler using SsoAdvanced™ Universal SYBR® Green Supermix. A 20 μl reaction mixture contained: 10 μl 2 × SYBR Green Master, 0.4 μl forward primer (10 μM), 0.4 μl reverse primer (10 μM), 2 μl cDNA, and 7.2 μl dd $H_2O$ in a 96-well plates. The amplification conditions were as follows: 95 °C for 3 min, 40 cycles of 95 °C for 5 s and 58 °C for 15 s. Melting curve was analyzed to determine primer specificity. 2−ΔCT method was used for calculating with reference gene RPL27. Primers used were: SLAMF7 (forward AGAACACAGAGTACGACACAAT/reverse CAGTGGAGTAAACCGT ATTTGC), RPL27 (forward CCGGACGCAAAGCTGTCATCG/reverse CTTGCCCATGGCAGCTGTCAC).

**In vitro macrophage phagocytosis assay.** Macrophages were harvested and pre-seeded at $1.5 \times 10^4$ cells/well in 96-well plates. Tumor cells were labeled with cell proliferation dye V450 (Thermofisher) or pHrodo green pH indicator (Thermofisher) according to manufacturer's instructions. Subsequently, tumor cells were incubated with or without anti-human CD47 IgG4 antibody (Inhibrix) or F(ab')2 of Inhibrix (both at 5 μg/ml) on ice for 1 h. Tumor cells were washed with cold PBS (two times) and added to pre-seeded macrophages (effector to target ratio of 1:5) and incubated for 3 h at 37 °C. Tumor cells were gently removed by washing with PBS 2–3 times and phagocytosis was analyzed by fluorescent microscopy (Leica, DM6000) or confocal microscopy (Leica SP8). For visualization of macrophages, residual cells were stained with anti-human CD11b-alexa594 (1 μg/ml) at RT for 45 min. The percentage of phagocytosis was calculated by counting the number of macrophages containing V450-labeled tumor cells per 100 macrophages. Each condition was quantified by evaluating three randomly chosen fields of view.

**Retrospective mRNA analysis of DLBCL patients.** Publicly available raw microarray expression data of DLBCL samples platforms (Affymetrix HG-U133A (GPL96) and Affymetrix HG-U133 Plus 2.0 (GPL570)) were extracted from the Gene Expression Omnibus (GEO)[20]. To identify DLBCL samples, the Simple Omnibus Format in Text (SOFT) files that contain metadata of each individual sample, were retained if they contained at least one of the keywords; DLBCL or DLCL. Next manually, pubmed identifiers pointing to relevant published manuscripts were used to confirm the samples represented de novo DLBCL. Cell lines and animal samples were excluded. Only rituximab, cyclophosphamide, doxorubicin, vincristine, and prednisone-treated DLBCL patient samples were used for further analysis. Principal component analysis (PCA) on the sample correlation matrix was used for quality control[21]. The first principal component (PCqc) of such an expression microarray correlation matrix describes nearly always a constant pattern that dominates the data. This first PCA explains 80–90% of the total variance, which is independent of the biological nature of the sample being profiled. The correlation of each individual microarray expression profile with this PCqc can be used to detect outliers, as arrays of lesser quality will have a lower correlation with the PCqc. We removed samples that had a correlation $R < 0.8$. Probe 213857_s_at (CD47 gene) and 219159_s_at (SLAMF7 gene) were used for the analyses. For patient characteristics, see Supplementary Tables 1 and 2. For GEO accession numbers, and distribution of SLAMF7 and CD47 mRNA expression see Supplementary Figure 2A–C.

**Estimated immune cell type fractions.** CIBERSORT was used to estimate the fraction of the three subtypes of macrophages (M0, M1, and M2). CIBERSORT is a method for characterizing cell composition of complex tissues from their GEP that has been shown to have strong agreement with ground truth assessments in bulk tumors. The 680 DLBCL GEP was used in combination with the leukocyte gene signature matrix, LM22, to distinguish 22 hematopoietic cell types, including the three subtypes of macrophages. For each sample (DLBCL GEP), the sum of all estimated hematopoietic cell-type fraction equals 1.

**Statistical analysis.** Effect of CD47 antibody on phagocytosis was determined by paired Student's t-test, based on a minimum of 3–5 different donors. Time-to-event data was analyzed using the Kaplan–Meier method and the log-rank test to compare the survival distributions between the different groups. OS was defined as the time from primary diagnosis to death from any case. Survivors were censored on the last date that they were known to be alive or when followed up longer than 6 years. Patients were sorted based on CD47 expression or SLAMF7 expression. The 25% of patients with lowest and highest expression were used to determine influence of expression on survival. All statistical analysis are tested two-sided and p-values < 0.05 were considered statistically significant. All analyses were conducted using SPSS statistics (version 23.0 Armonk, NY, IBM Corp.), STATA 14

(StataCorp LP, College Station, TX), and GraphPad Prism (GraphPad Prism [version 7.0b]; GraphPad Software, La Jolla, CA).

**Reporting summary**. Further information on experimental design is available in the Nature Research Reporting Summary linked to this article.

## Data availability

The raw microarray expression data of DLBCL samples platforms (Affymetrix HG-U133A 316 (GPL96) and Affymetrix HG-U133 Plus 2.0 (GPL570)) were extracted from the Gene Expression Omnibus (GEO) from the https://www.ncbi.nlm.nih.gov/geo/ website, using the GEO accession numbers listed in Supplementary Table 2. Patient characteristics are reported in Supplementary Table 1. Data were analysed as described in the Methods section. All the other data supporting the finding of this study are available within the article and its supplementary information or from the corresponding author upon reasonable request.

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

## Acknowledgements

The authors would like to acknowledge financial support from Dutch Cancer Society grants RUG 2009-4355, RUG2011-5206, RUG2012-5541, RUG2013-6209, RUG2014-6986, and RUG20157887 (to E.B.), a Bas Mulder grant from Alpe d'HuZes/Dutch Cancer Society (RUG 2013-5960), a grant from the Netherlands Organization for Scientific Research (NWO-VENI grant 916-16025) and a Mandema Stipendium (to R.S.N.F.), and a Bas Mulder grant of Alpe d'HuZes/Dutch Cancer Society (RUG 2014-6727) and a Mandema Stipendium (to T.v.M.).

## Author contributions

Y.H., R.B., V.R.W., M.J., H.L., M.B. designed and performed experiments and analyzed the data. E.A., R.F. and M.B. contributed intellectually in the study. E.B., T.v.M., G.H. designed the research, analyzed the data, wrote the manuscript. Y.H. wrote the manuscript, E.B. and T.v.M. supervised the entire study.

## Additional information

**Competing interests:** The authors declare no competing interests.

