## [Peer Review File · Nature Communications]

This manuscript has been previously reviewed at another journal that is not operating a transparent peer review scheme. This document only contains reviewer comments and rebuttal letters for versions considered at Nature Communications. Mentions of the other journal have been redacted.

Reviewers' Comments:

Reviewer #1:

Remarks to the Author:

In this work, the authors challenge the previously held notion that SLAMF7 expression on cancer cells is not required for phagocytosis after CD47:SIRPa blocking between targets and Phagocytes. The authors use a series of approaches to demonstrate these points – the data presented support their conclusions. The final set of analysis using patient derived data on CD47 expression versus SLMF7 expression further correlates with the authors' claims. As a reviewer who has read the earlier paper in Nature where this SLAMF7 homotypic interaction requirement was reported, and also now reading this work, it is impossible to directly compare the two sets of data as these were done in different labs across the Atlantic. Nevertheless, on its own, the data presented here is solid, and after all, it is part of the scientific discourse to repeat observations to ensure their validity and should be reproducible in different people's hands. Therefore, I recommend publication of this work in Nature Communications, and allow other investigators to further assess how best to move forward in whether or not to use the SLAMF7 expression as a marker in human therapeutic considerations.

Reviewer #2:

Remarks to the Author:

I am satisfied with the responses of the authors to my comments

The paper provides a provocative view on the controversial role of SLAMF7 in tumor phagocytosis that will stimulate debate and further studies